# TMEM97/Sigma 2 Receptor Increases Estrogen Receptor α Activity in Promoting Breast Cancer Cell Growth

**DOI:** 10.3390/cancers15235691

**Published:** 2023-12-02

**Authors:** Yuanqin Zhang, Xiangwei Fang, Jiuhui Wang, Daotai Nie

**Affiliations:** Department of Medical Microbiology, Immunobiology and Cell Biology, Southern Illinois University School of Medicine and Simmons Cancer Institute, Springfield, IL 62702, USA; yuanqin@wustl.edu (Y.Z.); xfang97@siumed.edu (X.F.); jwang29@siumed.edu (J.W.)

**Keywords:** estrogen receptor α, sigma receptors, TMEM97, mTOR, S6K1, tamoxifen

## Abstract

**Simple Summary:**

Selective estrogen receptor modulators are a major treatment modality for estrogen receptor-positive breast cancers, but the emergence of resistance or a lack of initial responsiveness is a major cause of treatment failures. This study describes TMEM97, also known as sigma 2 receptor, as a novel regulator of estrogen receptor activation. High expression of TMEM97 is found in breast tumor tissues with estrogen receptor positivity. Depletion of TMEM97 expression reduces estrogen receptor activities and breast cancer cell growth. Increased expression of TMEM97 in breast cancer cells stimulates estrogen receptor activities and growth. Further, TMEM97 increases the resistance of breast cancer cells to tamoxifen through elaborating the mTOR/S6K1 signaling. The results suggest an important role for TMEM97 in estrogen receptor activation and resistance to tamoxifen in breast cancer cells.

**Abstract:**

Aberrant estrogen receptor (ER) signaling is a major driver of breast tumor growth and progression. Sigma 2 receptor has long been implicated in breast carcinogenesis based on pharmacological studies, but its molecular identity had been elusive until TMEM97 was identified as the receptor. Herein, we report that the TMEM97/sigma 2 receptor is highly expressed in ER-positive breast tumors and its expression is strongly correlated with ERs and progesterone receptors (PRs) but not with HER2 status. High expression levels of TMEM97 are associated with reduced overall survival of patients. Breast cancer cells with increased expression of TMEM97 had a growth advantage over the control cells under both nutrition-limiting and sufficient conditions, while the knockdown of TMEM97 expression reduced tumor cell proliferations. When compared to their vector control cells, MCF7 and T47D cells with increased TMEM97 expression presented increased resistance to tamoxifen treatment and also grew better under estrogen-depleted conditions. The TMEM97/sigma 2 receptor enhanced the ERα transcriptional activities and increased the expression of genes responsive to estrogen treatment. Increased TMEM97 also stimulated the mTOR/S6K1 signaling pathways in the MCF7 and T47D cells. The increased level of active, phosphorylated ERα, and the enhanced resistance to tamoxifen treatment with increased TMEM97, could be blocked by an mTOR inhibitor. The knockdown of TMEM97 expression reduced the ERα and mTOR/S6K1 signaling activities, rendering the cells with an increased sensitivity to tamoxifen. The observations suggest that the TMEM97/sigma 2 receptor is a novel regulator of ERα activities in breast tumor cell growth.

## 1. Introduction

Breast cancer is one of the most common cancers in women. Aberrant estrogen receptor (ER) activity is the primary driver of tumor growth and progression in a majority of breast cancer patients. “Estrogen” refers to a family of female hormones, including estrone (E1), estradiol (E2), estriol (E3), and estetrol (E4) that can bind to ERs with different affinities, with E2 as the predominant circulating estrogen in humans [1]. The two isoforms of estrogen receptors, ERα and ERβ, are both members of nuclear receptors of transcriptional factors. The canonical cascade leading to ER transcriptional activity is initiated by the binding of estrogen, which results in a conformational change that induces receptor dimerization [2], and translocation into the nuclei. Then, the E2–ER complexes bind directly to the estrogen response elements (EREs) in the chromatin, the enhancer regions within or close to promoters, and/or the 3′-untranlated regions of the target genes [3,4]. Once bound to EREs, the E2–ER complexes can initiate gene expression by recruiting its specific co-regulatory proteins, known as co-activators and co-repressors [5]. In addition, there is also the indirect genomic activation of nuclear estrogen receptors and non-genomic signaling membrane receptors.

Selective estrogen receptor modulators (SERMs) compete with estradiol to bind to the ligand-binding domain of ERα [6]. Tamoxifen is one of the most-used SERMs in the treatment of breast cancer, which works as an antagonist in breast tissues but an agonist in the uterus [7]. It has been used as endocrine therapy for ER-positive metastatic breast cancers for over 40 years. However, about half of ER-positive breast cancer patients present resistance to tamoxifen, and further, nearly all initial responders will eventually acquire resistance. Therefore, it is important to elucidate how breast cancers acquire resistance to endocrine therapy and identify new therapeutic opportunities to improve the efficacy of endocrine therapy.

Sigma receptors, classified into sigma 1 (σ1) and sigma 2 (σ2) receptors [8,9], are found in a variety of tissues and organs. Sigma 2 receptor has been explored as a proliferation biomarker and imaging target for cancer diagnosis for years before transmembrane protein 97 (TMEM97, also known as MAC30) was identified as the *bona fide* σ2 receptor [10]. TMEM97 has been predominantly studied for its role in regulating cholesterol homeostasis [11]. Elevated expression of TMEM97 or MAC30 has been linked with poor clinical parameters in gastric, colorectal [12], breast [13], and ovarian [14] cancers, but the biological mechanism involved remains unknown. Herein, we report that the TMEM97/σ2 receptor modulates the growth of ER-positive breast cancer cells via regulating ERα activities.

## 2. Materials and Methods

### 2.1. Chemicals

17 β-Estradiol (E2), (*Z*)-4-Hydroxytamoxifen (OH-Tam), ICI 182,780 (Fulvestrant), and rapamycin were purchased from AdooQ Bioscience. Cycloheximide and other common chemicals were purchased from Sigma-Aldrich (St. Louis, MO, USA).

### 2.2. Immunohistochemistry

Immunohistochemical analysis on a paraffin-embedded breast tissue array was carried out using commercial validated anti-TMEM97 rabbit polyclonal antibody (1:100) (Thermo-Fisher Scientific (Waltham, MA, USA)). The breast cancer tissue microarray (BC081116b, US Biomax, Inc. (Derwood, MD, USA)) contained 110 samples with 100 cases of breast invasive ductal carcinoma and 10 cases of adjacent breast tissue as a control with a single core per case. Each case had pathology information for ER, PR and HER2, TNM, clinical stage and grade. Out of these total 110 samples, 102 cases had a valid ER status, the remaining 8 cases had an undetermined ER status. Similarly, 102 cases had a valid PR status, and 101 cases had a valid HER2 status. For IHC staining, the tissue microarray slides were deparaffinized, rehydrated, and the antigen retrieved by placing in Declere working solution (Cell Marque (Rocklin, CA, USA)) in an electric pressure cooker for 30 min. After a hot rinse with boiling Declere, slides were cooled for 5 min. Then, the slides were washed with distilled water (3 changes), dehydrated through 95% and 100% alcohol (2 changes), and cleared in xylene (2 changes). Slides were processed for immunohistochemical staining using the Histostain Plus Broad Spectrum kit (Zymed (Oxnard, CA, USA)) following the manufacturer’s instructions. After mounting with DPX Mount (Sigma (St. Louis, MO, USA)), slides were examined and photographed under microscopy (OLYMPUS DP73). The degree of staining was calculated by H-Score using the following formula: H-Score = 100 × (% at 0) × 0 + (% at 1+) × 1 + (% at 2+) × 2 + (% at 3+) × 3].

### 2.3. Cell Culture and Transfections

MCF7 and T47D, two estrogen receptor-positive breast cancer cell lines, were purchased directly from ATCC and used within six months. MCF7 cells were grown and maintained in DMEM/High glucose with 4.0 mM L-glutamine and sodium pyruvate medium (HyClone (Utah, UK)). T47D cells were grown and maintained in RPMI 1640 medium with L-glutamine (HyClone). All cell culture media were supplemented with 10% fetal bovine serum (ATCC) and Antibiotics-Antimycotics (100 units/mL penicillin, 100 μg/mL streptomycin, and 250 μg/mL amphotericin B), and cultured in a humidified incubator under an atmosphere containing 5% CO_2,_ at 37 °C.

For estrogen- and serum-deprived conditions, cells were seeded in full culture medium (10% FBS) overnight and changed to phenol red-free, DMEM/High glucose medium with L-glutamine, and sodium pyruvate, supplemented with 0.5% charcoal-stripped serum for 24 h before indicated treatments.

### 2.4. Cloning and Expression of TMEM97

The ORF for the human TMEM97 gene (ENSG00000109084) was amplified from breast cancer MCF7 cells using Phusion High-Fidelity DNA Polymerase (Thermo Scientific) according to the manufacturer’s protocol and a nested PCR technique and cloned into a pCDH-myc vector. GFP gene was directly linked to the C-terminal end of TMEM97 and inserted into the same pCDH-myc vector (Appendix A). Meanwhile, the CopGFP gene of the pCDH-myc vector was deleted at the same time. Correct insertion of the TMEM97-EGFP was confirmed via DNA sequencing.

For TMEM97-EGFP overexpression, 293T cells were infected with lentiviral vector and pCDH-TMEM97-EGFP along with packaging vectors (System Biosciences (Palo Alto, CA, USA)) by using DNAfectin^TM^ liposome transfection reagent. Viral supernatants were collected 48 h after transfection, diluted 1:1 in fresh medium in the presence of polybrene (4 μg/mL), and added to MCF7 and T47D cells. The empty pCDH vector was used as a control. After 48 h of infection, cells with green fluoresces were sorted out via flow cytometry, and maintained in full culture conditions.

### 2.5. Knockdown of TMEM97 Expression

For TMEM97 shRNA stable knockdown, TMEM97 shRNA constructs were purchased from Dharmacon (Appendix A). Three different constructs, 131 (Clone ID: V3LHS_333131), 835 (Clone ID: V2LHS_54836), 602 (Clone ID: V3LHS_404602), were used with different targeting regions. Sequencing primer 5′-GCATTAAAGCAGCGTATC-3′ was used for all three of the constructs. The empty vector pGIPZ was used as the control. The production of lentiviral particles and transduction were the same as described above for lentiviral overexpression. The GFP-positive transduced cells were selected via cell sorting and maintained in growth medium for expansion and characterizations.

### 2.6. Colony Formation Assay

A total of 100 MCF7 (TMEM97/PCDH) cells per well were seeded into tissue culture 12-well plates. They were cultured in 5% FBS DMEM or 0.5% charcoal-stripped FBS phenol red-free DMEM culture medium. Media were replaced weekly. Subsequently, the cells were washed with PBS then fixed in 4% formaldehyde. The fixed cells were stained with crystal violet (Fisher Scientific), washed with water, and allowed to air-dry. Colonies were counted and dishes were photographed.

### 2.7. Luciferase Assay

A total of 1 × 10^5^ cells per well were seeded in 24-well culture plates with full culture medium overnight. Next day, cells were transiently transfected with ERE-TATA-Luc reporter construct and Renilla construct in a 10:1 ratio for 24 h. Then, the culture medium was changed to 0.5% CS-FBS phenol red-free medium for another 24 h prior to E2 treatment for 5 h. Cells were harvested for dual luciferase assay afterwards. The luciferase activities were measured by dual luciferase assay kit (Promega (Madison, WI, USA)) following the instructions described by the manufacturer and normalized with the Renilla activities. Each experiment was performed with 4 repeats per sample.

### 2.8. MTS Assay

A total of 1 × 10^4^ cells were seeded into each well of 96-well plates overnight. The next day, cells were changed to the desired culture media with or without compound treatment. After 4 days of culturing, the medium was removed and 1 × PBS mixed with 20 µL MTS reagent (Promega) was added for each well and incubated for 1 h. Absorbance of 490 nm light (OD_490nm_) was determined using a spectrophotometer. Each data point was represented as the mean of 4 repeats.

### 2.9. RNA Isolation and qRT-PCR Analysis

Total RNAs were isolated using GeneJET RNA Purification Kit (Thermos Scientific) following the manufacturer’s instructions. An amount of 1 μg of RNA was used for reverse transcription performed via a ProtoScript^®^ II First Strand cDNA Synthesis Kit (New England Biolabs (Ipswich, MA, USA)). The cDNA products were used as templates for real-time qRT-PCR with SYBR green qPCR mixture by AzuraQuant Green Fast qPCR Mix LoRox following the manufacturer’s protocol. Reactions were run on the 7500 Real-Time PCR System (Applied Biosystems (Waltham, MA, USA)) and the reactions were performed in triplicate. Data were normalized to β-actin and relative mRNA expression was determined using the ΔΔCt. Gene-specific primer sequences are listed in Table 1.

### 2.10. Immunoblots and Antibodies

Cells were lysed with 2× SDS lysis buffer containing a protease inhibitor cocktail on ice. The cell lysate was sonicated for 10 s, and was then followed by heating at 95 °C for 5 min. All samples were loaded at the same amount and separated by SDS-PAGE gel in a Bio-Rad Protean II system. After transferring proteins to a polyvinylidene difluoride (PVDF) membrane, the membrane was blocked with 5% BSA for 60 min at room temperature and incubated with the primary antibody at appropriate dilutions in 5% BSA at 4 °C. After overnight incubation with appropriate primary antibodies, the membrane was washed three times with Tris-buffered saline (TBS) containing 0.1% Tween for 5 min each time and probed with fluorescently labeled secondary anybody (1:10,000) for 1 h at room temperature. The membrane was then washed three more times with TBS-T for a total of 15 min. The immunoblots were visualized by Odyssey Infrared Imaging. Densitometry was performed using the Odyssey 2.0 infrared imaging system.

The following antibodies used in this study were purchased from Cell Signaling. Rabbit monoclonal antibodies: anti-β-actin, anti-GAPDH, anti-p-ERα (Ser167), anti-ERα, anti-p-S6K1 (Thr389), anti-S6K1, anti-p-S6 (S236/S236), anti-p-S6 (S240/S244), anti-p-Erk1/2, anti-p-AKT, anti-p-c-Raf, anti-p-P38 MAPK, with the dilution of 1: 1000 for Western blot. TMEM97 polyclonal antibody was purchased from Thermo-Fisher Scientific with the dilution of 1:100–1:250 for Western blot.

### 2.11. Statistical Analysis

All data were analyzed using spreadsheets with Microsoft Excel 2019 or GraphPad Prism V (GraphPad, San Diego, CA, USA). Data were presented as mean ± standard error of the mean (SEM). Statistical comparisons were made using unpaired, two-tailed Student’s *t*-test. Differences were considered statistically significant if *p* ≤ 0.05.

## 3. Results

### 3.1. Increased TMEM97 Protein Expression Is Associated with Estrogen Receptor Status

With TMEM97 identified as σ2R, we first determined the expression pattern of TMEM97 in human breast cancer tissues via immunohistochemical staining (IHC) with a validated TMEM97 antibody. The positive staining of TMEM97 was quantified under a microscope using a double-blinded approach and the H-scores calculated. It was found that the intensity of the TMEM97 signaling was progressively higher with a higher estrogen receptor (ER) status, with the most intensive staining in the ER +++ tissue samples and it reduced gradually in the ER ++, ER +, and ER negative samples (Figure 1A). As shown in Figure 1A and Table 2, breast cancer patient tissue samples with the highest ER expression (ER+++) (n = 42, 42%) exhibited the highest H-score of 123.16 ± 8.70, and the H-scores reduced as the ER-positive status declined, with the ER ++ samples (n = 16, 15.7%) scoring 77.93 ± 14.97 and ER + samples (n = 15, 14.7%) scoring 52.50 ± 10.06. The differences between the ER +++ group and all the others are statistically significant with *p* values less than 0.05.

Interestingly, we observed similar results linking TMEM97 with the PR status similar to the ER status. TMEM97 staining is predominantly strong in the PR-positive tissue samples, with a statistically significant expression in the PR +++ tissue samples (H-score = 119.89 ± 8.81, n = 23, 22.5%) (Figure 1B and Table 2). In contrast to ER (Figure 1C and Table 2) or PR (Figure 1D and Table 2), the HER2 status did not lead to any different expression pattern of TMEM97 (Figure 1E and Table 2).

To further assess the clinical relevance of the TMEM97/σ2 receptor in breast cancer, we queried TMEM97 using a public database, Kaplan–Meier plotter [15]. This database uses aggregate microarray data to analyze the prognostic value of a specific gene by dividing the cohorts into two groups (High refers to the upper quartile and Low refers to the lower quartile) according to the gene expression. Multiple probes are available for a particular gene to make the database more vigorous. There are four different probes for the TMEM97 gene. As shown in Figure 1F, a high level of TMEM97, as detected by three out of the four probes, was significantly associated with a low relapse-free survival (RFS) in all types of breast cancers. The highest hazard ratio (HR) is 1.5 (*p* < 0.001), as revealed by the probe set 212279_at, then followed by the probe 2122811_s_at and 212282_at with HR 1.4 (*p* < 0.001) and 1.36 (*p* < 0.001), respectively. Remarkably, if we gated the two cohorts with estrogen receptor-positive patients (Figure 1G), all four probe sets exhibited a significant correlation of a high TMEM97 mRNA expression with a poor relapse-free survival. Among them, the highest HR is 1.61 (*p* < 0.001), as detected by probe 212281_s_at, and the lowest HR is 1.32 (*p* < 0.01) by 214283_at. The bioinformatics data suggest that the survival of breast cancer patients is impacted by the TMEM97 mRNA level, especially for ER-positive patients.

### 3.2. Stimulation of ER-Positive Breast Cancer Cell Growth and Survival under Different Culture Conditions by TMEM97 Overexpression

To determine the biological functions of TMEM97, we cloned and constructed the TMEM97 overexpression plasmid using the pCDH-CMV lentivector (illustrated in Appendix A). Then, we increased the expression of TMEM97 in the MCF7 and T47D cells as they are human breast cancer cell lines with estrogen and progesterone receptors (ER-positive breast cancer cells) by lentivirus infection, with GFP as an indication of successful transductions and predominant localization of TMEM97 in the cytoplasm (Appendix A).

The overexpression of TMEM97 is confirmed at both the protein (Figure 2A) and mRNA levels (Figure 2B). The molecular weight of the TMEM97 protein itself is around 24 kDa, and the molecular size of the fusion protein TMEM97-EGFP would be around 45 kDa. As expected, there were two bands (~45 KDa and ~24 KDa) showing in the overexpression cells, while there was only one band in the PCDH control cells (Figure 2A). Next, we performed quantitative PCR using three different sets of primers targeting three different positions of the TMEM97 gene from the primer bank [16] (Figure 2B).

To determine whether the increased expression of TMEM97 could promote and stimulate breast cancer cell growth and survival, we performed the proliferation study via MTS assay under different culture conditions and found that the overexpression cells (TM) tend to survive better than the control cells (PCDH) under serum-depleted conditions, though the differences were not significant, as shown in Figure 2C. Then, we gradually increased the serum levels in the culture media, the growth diverged at Day 4 for the MCF7 cells cultured at 2% FBS (Figure 2D), and became more significant under 5% FBS culture conditions on Day 3 and Day 4 for both the MCF7 and T47D cells (Figure 2E). On Day 4, at 5% FBS, the overexpression cells grew at 1.31-folds and 1.36-folds faster than the control cells in MCF7 and T47D, respectively (Figure 2E). If the cells were cultured under normal conditions, 10% FBS, the growth advantage via TMEM97 overexpression even started at Day 2, 1.2-folds for MCF7 and 1.45-folds for T47D. And the trend kept the same to Day 4, 1.24-folds for MCF7 and 1.33-folds for T47D (Figure 2F).

### 3.3. TMEM97 Supports ER-Positive Breast Tumor Cell Growth and Colony Formation under Hormone Depletion

Phenol red, the widely used pH indicator in tissue culture medium, has been found to be structurally similar to some nonsteroidal estrogens and exhibits significantly estrogenic activities. The concentration of phenol red presented in culture medium is relatively high so that it could stimulate the cell growth of some estrogen-sensitive cells [17,18,19], especially for the human breast cancer-derived MCF7 and T47D cells [17]. In addition to the phenol red, another interference in the study of the estrogen-responsive cell system is the estrogen that already exists in FBS, since this hormone could pass through the placenta to the fetus’ circulation. To study the effects of steroid hormones in vitro without the interference from the growth environment, a “hormone-free” condition is needed, so dextran-coated charcoal was utilized to deplete the hormone from the FBS. Charcoal-stripped fetal bovine serum was produced by absorbing with activated carbon that removes non-polar material such as lipophilic (lipid-related) materials, like a virus, certain growth factors, hormones, and cytokines, but has little effect on salts, glucose, amino acids, etc. The phenol red-free medium supplied with charcoal-stripped FBS could keep the cells in estrogen free media for testing [19].

Since TMEM97 has been found to be positively associated with estrogen receptors, to understand their relationships without the interference of surrounding estrogen, we changed the culture medium to phenol red-free medium supplied with charcoal-stripped FBS (CS-FBS) at different concentrations. And we found that even when supplied with a very low concentration of CS-FBS (0.5%), the TMEM97 overexpression cells still showed an enhanced proliferation in both the MCF7 and T47D cells (Figure 3A). With an increased concentration of CS-FBS (5%), as shown in Figure 3B, the difference between the overexpression and the control started to show even at Day 2. The data suggest that TMEM97 could also promote cell growth under estrogen-depleted conditions for ER-positive breast cancer cells.

Next, we determined whether TMEM97 promotes the colony formation of breast tumor cells. A total of 100 MCF7 (TMEM97/PCDH) cells were seeded into each well in a tissue culture plate and then cultured in 5% FBS DMEM or 0.5% charcoal-stripped FBS phenol red-free DMEM culture medium. After 13 days of culturing in 5% FBS DMEM media, the MCF7 TMEM97 showed increased colony formation when compared with the pCDH cells (Figure 3C). Under culture conditions with 0.5% charcoal-stripped FBS phenol red-free DMEM culture media, the pCDH cells hardly formed colonies after 13 days, but the TMEM97 cells still showed a strong ability to form colonies (Figure 3D). The results for the group in the 0.5% charcoal-stripped FBS phenol red-free DMEM culture media after 19 days showed a similar pattern (Figure 3D). The results suggest that TMEM97 could promote cell growth and proliferation even under estrogen-depleted conditions for ER-positive breast cancer cells.

### 3.4. Inhibition of ER-Positive Breast Cancer Cell Growth and Survival by TMEM97 Knockdown

To determine whether TMEM97 is required for breast tumor growth, we performed shRNA-mediated TMEM97 knockdown experiments on the MCF7 cells. Two of the three shRNA target the 3′ UTR of TMEM97 while one shRNA targets the ORF of TMEM97, as shown in Appendix A. The efficiency of the knockdown was tested at the protein level (Figure 4A). Then, at the mRNA expression level, all three knockdown cell lines had reduced the TMEM97 expression level to about 30% to 40% of the control, especially the 836 and 602 cells, as confirmed by three different primers (same primers as used in the overexpression confirmation) (Figure 4B).

Next, we performed the cell growth assay for the shRNA-mediated knockdown cells under different culture conditions. As expected, the knockdown cells showed a significantly reduced cell growth, especially in the cells cultured under FBS-free conditions, which was around an 80% reduction on Day 2, and 60% on both Day 3 and Day 4 for all three knockdown cells (Figure 4C). However, with a continuously increased percentage of FBS supplement, the growth disadvantage was, finally, eliminated (Figure 4F). Notably, when cultured in 2% FBS, the decreased cell growth still existed for all the knockdown cells on Day 3. On Day 4, the difference only remained in 836 and 602 knockdown cells, but not for 131 cells (Figure 4D). Similar results were observed in Figure 4E after 4 days of 5% FBS cultivation. A different cell growth reduction between the 131, 836, and 602 cells could be due to a different efficiency of the TMEM97 knockdown. As indicated in Figure 4B, the 836 and 602 cells showed a more reduced TMEM97 mRNA expression compared to the 131 cells.

We also performed the proliferation assay for the TMEM97 knockdown cells under estrogen-depleted conditions. We found that the knockdown cells presented reinforced growth suppression under estrogen-depleted conditions, regardless of what percentage of charcoal-stripped FBS was provided (Figure 4G,H). Consistent with previous data, the inhibition effects were more pronounced in the 836 and 602 cells. As shown in Figure 4G,H, the cell proliferation rate for the 602 cells remained below 50% of the control cells after 4 days of cultivation both in the 0.5% CS-FBS and 5% CS-FBS.

These data suggest that the knocking down of TMEM97 expression could suppress breast cancer cell growth under starvation culture conditions, or estrogen deprivation conditions.

### 3.5. Estrogen Receptor Transcriptional Activity Is Upregulated via the Overexpression of TMEM97/σ2 Receptor

Based on the association of TMEM97 expression with estrogen receptors and the increased proliferation of ER-positive breast cancer MCF7 cells, we determined whether TMEM97 regulates ER transcriptional activity. Firstly, we used qRT-PCR to evaluate changes at the transcript level of the ERα target genes *NRIP1* [20], *ABCA3* [21], *GREB1* [22], and *ALOX12B.* Among them, *NRIP1, ABCA3,* and *GREB1* are ERα upregulated genes, and *ALOX12B* is an ERα downregulated gene. Cells were cultured in serum-starved and estrogen-depleted conditions (0.5% CS-FBS phenol red-free medium) for 24 h before treatment with DMSO (Ctrl) and 17β-estradiol (E2) at 10 µM for 5 h.

We found a significant induction of *NRIP1* and *ABCA3* in the TMEM97 overexpression cells with or without the E2 treatment (Figure 5A,B). The significant induction of the *GREB1* transcript level only existed without the E2 treatment (Figure 5C). For the ERα downregulated gene *ALOX12B*, the transcript level was reduced in the overexpression cells with or without the E2 treatment (Figure 5D). These data suggest that TMEM97 itself could modulate the expression of ERα target genes, with or without E2 stimulation. No differences observed for GREB1 after the E2 treatment may be due to the super-sensitive responses for this gene. Almost all (~95%) the GREB1 binding region is shared by estrogen receptor binding, so that it is the most estrogen-enriched ER interactor [22].

To measure the direct effect of TMEM97 on the ERα transcriptional activity, we conducted a luciferase-based ERα reporter assay. As indicated in Figure 5E, when transfected with ERα, E2 increased the ERE activity by around four-folds after 5 h of treatment. When co-transfected with TMEM97-EGFP, the transcriptional activities after the E2 treatment increased to six-folds, and 1.8-folds relative to the ERα vector alone. Next, we tested the ERα transcriptional activities in the MCF7 stable TMEM97 overexpression cells under the same culture conditions (Figure 5F), and observed that after the E2 treatment, the induced transcription in the TMEM97 overexpression cells was significantly higher than in the control cells (>1.7-folds). No transcriptional activity differences were observed in the absence of E2 in the transiently transfected cells or in the stable overexpression cells (Figure 5E,F). The data suggest that the increased TMEM97 expressions in the MCF7 cells led to increased ERα activities in response to E2.

### 3.6. TMEM97/σ2 Receptor Activates ERα Independent of Estrogen and Contributes to Tamoxifen Resistance

To determine the mechanism involved in TMEM97 regulating ER activities, we investigated the influence of TMEM97 overexpression on ERα protein. Serine 167 of ERα has been identified as the major estrogen-induced phosphorylation site in breast cancer cells [23]. We found that ERα p-S167 was markedly increased in an estrogen-dependent manner in the MCF7 cells (Figure 6A). However, in the TMEM97 overexpression cells, the differences in p-S167 before and after the E2 treatment were not obvious, because the basal level of p-S167 was extremely high in the overexpression cells even in the absence of E2 compared to the control cells (Figure 6A), suggesting that TMEM97 itself played a similar role to estrogen that could continuously activate ERα at its S167 site without the presence of estradiol.

Serine 167 of ERα has been studied for many years. Phosphorylation of this site could activate and promote ERα-dependent transcription and cellular proliferation and is attributed to increased resistance to tamoxifen treatment [24,25,26]. Various studies have shown that increased Ser167 phosphorylation correlates with a poor prognosis in different cancer types [27,28]. Tamoxifen is widely used as a chemotherapeutic and preventative medicine for estrogen receptor-positive tumors [29].

To elucidate whether the sustained activated ERα S167 level in the TMEM97 overexpression cells would lead to tamoxifen resistance, we performed a proliferation assay with treatment of 4-hydroxytamoxifen (OH-Tam), a metabolite of tamoxifen with greater potency. The MCF7 cells were deprived of E2 and then stimulated with E2 in the presence of a different concentration of OH-Tam. After 4 days of treatment, PCDH and TMEM97 showed significant differences in the response to the OH-Tam treatment, as shown in Figure 6B. The inhibition effects of OH-Tam in the PCDH cells could be observed at 1 nM. When increased to 1 µM, the percentage of viable cells dropped to less than 50%. While in the TMEM97 overexpression cells, OH-Tam exhibited a weak agonist effect on cell proliferation at a low concentration (0.1 nM and 1 nM). And when treated with 1 µM for 4 days, the viable cells still remained around 70%, suggesting that TMEM97 overexpression cells are more resistant to tamoxifen treatment.

### 3.7. Activation of ERα by TMEM97 Is through mTOR/p70S6K1 Signaling Pathway and Can Be Blocked by mTOR Inhibitor

To further investigate whether the endocrine resistance mechanism is due to the persistent stimulation of ERα Ser167, we examined the potential signaling pathway through which active p-ERα (S167) was increased in the MCF7 TMEM97 overexpression cells. Ser167 can be phosphorylated by ERK1/2/MAPK [30,31], AKT [32], p90 ribosome S6 kinase (p90RSK) [24], CK2 [23], and mTOR/p70S6K1 [33,34]. So, several Western blots were performed from the signaling pathways mentioned above to reveal the possible cascades that participated in the TMEM97-mediated ERα phosphorylation. We found that the mTOR/p70S6K1 signaling pathway was the only one that changed dramatically in the overexpression cells, but not others (Figure 6C). Similar to p-ERα S167, when cells were deprived of estrogen for 24 h, the endogenous p70S6K1 protein remained active through TMEM97 overexpression compared to the control cells, suggesting the involvement of 40S ribosome S6 kinase 1 [35] (Figure 6C).

The mTOR pathway has been identified as a key regulator of cell growth and proliferation in responses to extracellular stimuli, such as nutrition and growth factor availability. Rapamycin is a very well-known naturally derived mTOR inhibitor which can inhibit tumor cell proliferation and has immunosuppressive properties [36]. The 40S ribosomal S6 kinase 1 (S6K1) is one of the best characterized downstream targets of mTOR, which can rapidly respond to rapamycin treatment, resulting in dephosphorylation and inactivation [37]. To test if the sustained activation of p-ERα S167 by TMEM97 was through the mTOR/S6K1 and could be regulated by this signaling pathway, we evaluated the effects of rapamycin on protein phosphorylation at different concentrations in the absence of estrogen. Firstly, we could see the huge differences that both p-ERα S167 and p-S6K1 T389 were highly active in the TMEM97 overexpression cells without treatment (Figure 6D). But after the rapamycin treatment, even at 1nM, the phosphorylation levels of p-ERα and p-S6K1 were reduced to the similar levels of the control cells, especially for p-S6K1 (Figure 6D), indicating that the mTOR inhibitor could block the TMEM97 activation effect on S6K1.

### 3.8. TMEM97 Knockdown Reduces Both ERα and mTOR/S6K1 Signaling Activity and Increases the Tamoxifen Sensitivity

In parallel, we determined the effect of the TMEM97 expression knockdown on the ERα activity. Firstly, we analyzed the ERα transcription activity in the MCF7 cells with the TMEM97 expression knocked down under the same culture conditions as above. As shown in Figure 7A, the TMEM97 knockdown did not alter the endogenous ERα transcriptional activity before the E2 treatment. However, the TMEM97 knockdown significantly reduced, but not abolished, the stimulation of ERE activities by E2, from a 5.18-folds induction in the vector control cells, to 4.3-, 3.8-, and 3.7-folds in the 131, 836, and 602 cells, respectively (Figure 7A).

Next, we determined whether the knockdown of TMEM97 expression would affect the ERα and mTOR/S6K1 signaling. The GIPZ, 131, 836, and 602 cells were serum and estrogen deprived for 24 h and then stimulated with E2 for 5 h, and the cells were then harvested for immunoblotting for the analyses of the ERα phosphorylation. As shown in Figure 7B, p-ERα S167 could be activated by E2 stimulation in all cells, while without the stimulation, the endogenous activity of p-ERα S167 was suppressed in the TMEM97 knockdown cells, 836, and in particular, 602 cells. In addition to the reduction in the p-ERα levels, the mTOR/S6K1 signaling was suppressed (Figure 7C). The observations suggest that a high expression of TMEM97 in ER-positive breast cancer cells could enhance the ERα activity through the activation of the mTOR/S6K1 signaling pathway, while lacking TMEM97 expression would suppress the ERα activity together with the inhibition of mTOR/S6K1 signaling.

Since hyper-activation of p-ERα and p-S6K1 can lead to tamoxifen resistance, we next determined the response of the TMEM97 knockdown cells to tamoxifen. As shown in Figure 7D, all the TMEM97 knockdown cell lines showed significant tamoxifen sensitivity when compared to the vector control cells, under stimulation by E2. Apparently, under very low concentrations, the growth capacity of the TMEM97 knockdown cells was around 20–30% less than the control cells. Taken together, the knockdown of TMEM97 expression reduced the ERα and mTOR/S6K1 signaling activities, rendering the cells with an increased sensitivity to tamoxifen.

## 4. Discussion

In this study, we have made the following observations regarding the TMEM97/σ2 receptor in breast cancer. Firstly, TMEM97/σ2 receptor expression is notably elevated in breast tumors with a high expression of ERα and PR, while no correlation with the HER2 status is found. Secondly, theTMEM97/σ2 receptor modulates tumor cell proliferation and growth, especially when cells are in a nutrition-limited or estrogen-depleted condition. Thirdly, the TMEM97/σ2 receptor regulates the ERα activities in the breast cancer MCF7 and T47D cells. Fourthly, the TMEM97/σ2 receptor regulates the mTOR/S6K1 signaling pathways, rendering ERα with an increased level of active, phosphorylated ERα that can be blocked with an mTOR inhibitor. Fifthly, increased TMEM97/σ2 receptor expression renders MCF7 cells with an enhanced resistance to tamoxifen. These observations suggest that the TMEM97/σ2 receptor participates in breast tumor cell growth driven by estrogen receptor signaling and further renders tumor cells with an increased resistance toward endocrine therapeutics such as tamoxifen.

The clinical relevance of the TMEM97/σ2 receptor in breast cancer is at first indicated by the in silico analyses of a pubic database. In the database with tumor microarray data (Kaplan–Meier Plotter), breast cancer patients with high levels of TMEM97 mRNA had a significantly reduced overall survival as revealed by three out of four probes mapped to TMEM97. In ER-positive breast cancer, all four probes showed a significantly reduced survival in patients with high TMEM97 mRNA levels. While the in silico analyses suggest the potential involvement of TMEM97 in patient survival, cautions are warranted since the analyses only reveal a correlative, not causal, relationship between TMEM97 and patient survival.

A majority of breast cancers (70%) are ERα-positive, in which tumor growth and progression requires continued ERα activity and signaling. We found that TMEM97 could promote MCF7 breast cancer cell proliferation especially under nutrition-limited or estrogen-depleted conditions when compared with the vector control cells. On the other hand, the knockdown of TMEM97 reduced MCF7 cell growth. Similar results were observed in the T47D cells, another ER-positive breast cancer cell line. The data suggest that TMEM97 modulates the growth of ER-positive breast cancer cells.

There are several lines of evidence suggesting the regulation of ER activities by TMEM97. Firstly, TMEM97 overexpression increased the ER transcriptional activities as revealed by the reporter gene analyses in the MCF7 cells and co-transfection analyses. Secondly, TMEM97 knockdown reduced the ability of estrogen to enhance the ER transcriptional activities. Thirdly, TMEM97 overexpression can further increase the expression of ER target genes, and conversely, TMEM97 knockdown downregulated the expression of ER target genes. Further, we found that TMEM97 could regulate the ERα activity through or partially through the mTOR/S6K1 signaling pathway. When subjected to the mTOR inhibitor, the enhanced ERα signaling activity by TMEM97 was abolished. Taken together, these data lead to a conclusion that TMEM97 can modulate ER activities through modulating the mTOR/S6K1 signaling pathway.

For patients with ERα-positive breast cancer, selective estrogen receptor modulators (SERMs), such as tamoxifen, are the standard endocrine therapy, but resistance remains the major obstacle. About one-third of patients are resistant to tamoxifen at the beginning of treatment. Further, most patients who initially respond to tamoxifen will later develop resistance. In our study, we found overexpression of TMEM97 could lead to a sustained increase in the phosphorylation of ERα at Serine 167 and cells are more resistant to OH-tamoxifen treatment. Co-treatment with rapamycin could re-sensitize the TMEM97 overexpression cells to OH-tamoxifen. When TMEM97 expression is knocked down, both the ERα phosphorylation at Serine 167 and mTOR/S6K1 signaling activities are reduced. Tumor cells with TMEM97 knockdown were more sensitive to OH-tamoxifen than the vector control cells.

The mTOR itself, as a core component of both complexes, functions as a serine/threonine protein kinase regulating various cellular functions. The p70S6 kinase 1 is a downstream target of mTOR signaling, and phosphorylation of p70S6K1 at threonine 389 has been used as a hallmark of activation by mTOR [38,39], which then stimulates the initiation of protein synthesis through the activation of the S6 ribosomal protein [40]. Studies have supported that the hyperactivation of the mTORC1 signaling pathway contributes to endocrine resistance, which can be reversed by the mTORC1 inhibitor Everolimus in vitro [41,42,43]. And S6K1 could directly phosphorylate ERα specially at S167, leading to ligand-independent activation [33,34]. Subsequent studies showed that S6K1 and ERα constitute a positive feed-forward loop, the phosphorylated ERα by S6K1, which in turn promotes the transcription of RPS6KB1 to further mediate breast cancer cell proliferation [44]. In this study, increased S6K1 phosphorylation was consistently observed in the TMEM97 overexpressing cells. However, in the mTOR signaling pathway upstream of S6K1, no significant differences in the active Raf, Erk1/2, p38MAPK, or Akt were found between the TMEM97 overexpressing cells and the pCDH vector controls, although estrogen can activate them. However, the increased pS6K1 in the TMEM97 overexpressing cells is dependent on the mTOR1 activities since rapamycin could abolish the observed increase in pS6K1. Further studies are needed to determine the exact mechanism for TMEM97 to activate p70S6K1.

The mTORC1 blockers (rapalogs) have been evaluated in clinical trials and been used as a combination treatment with endocrine therapy for breast cancer and have achieved promising results [45,46,47,48,49,50,51]. In our study, we demonstrated that the overexpression of TMEM97 in breast cancer cells contributed to resistance to tamoxifen treatment, and the overexpressed cells continuously activated the mTOR/S6K1 signaling pathway and the ERα activity. The resistance and stimulation effects by TMEM97 could be blocked by the utilization of the naturally derived mTOR inhibitor rapamycin, suggesting that TMEM97 may activate the ERα through or partly through the mTOR signaling pathway independent of estrogen. Further studies are needed to evaluate the effects of more mTOR inhibitors especially mTORC1 inhibitors, like Everolimus, on TMEM97 overexpression cells both in vitro and in vivo.

Our present studies have some limitations. With TMEM97 identified as the bona fide σ2 receptor [10], our present study has not addressed whether the ligands or antagonists of this receptor can modulate ERα activities. There are many compounds that can bind, activate, or inhibit the σ2 receptor with various affinities and specificities, and it will be interesting to see how they can affect ER activities. However, since most of those small molecules often have off-target effects besides engaging the σ2 receptor, future studies are needed.

## 5. Conclusions

In conclusion, the TMEM97/σ2 receptor has been identified as a novel regulator of ERα in breast cancer cell growth and responses toward tamoxifen. As shown in Figure 7E, TMEM97 can regulate the activities of ERα through modulating the ERα binding to estradiol and binding to responsive elements in the promoters of ER target genes. Another mechanism involved is the TMEM97 modulation of the cellular signaling, particularly the mTOR/S6K1 signaling, in the activation of ERα. Further studies are needed to determine whether TMEM97 can be a valid target of intervention to modulate ER activities and to reduce resistance toward endocrine therapy.

## Figures and Tables

**Figure 1 cancers-15-05691-f001:**
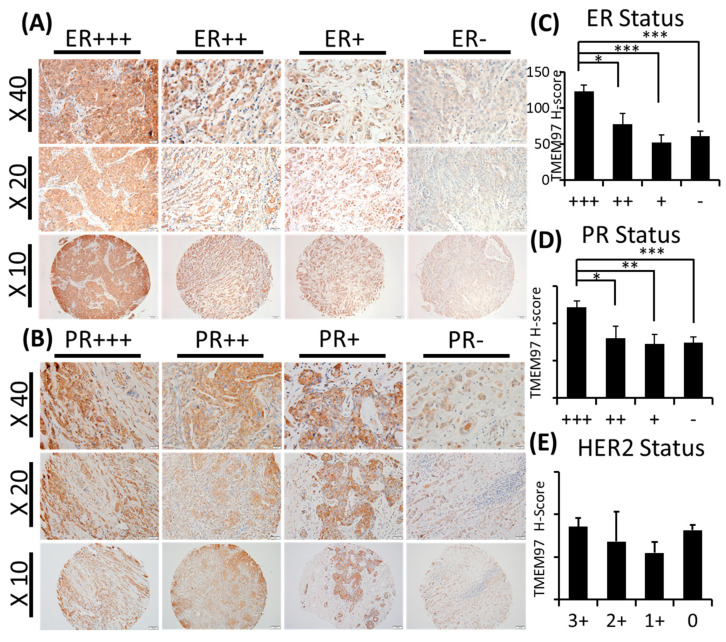
TMEM97/ σ2 receptor protein expression in breast cancer tumor tissues and its association with ER, PR, and patient survival. (**A**,**B**): IHC staining of TMEM97 in breast cancer tissue microarrays. Representative microscope images displayed as the ER status (**A**) or PR status (**B**) from strongest expression (+++) to negative. Bottom panel image at 10x with the medium panel image at 20× and the top panel image at 40×. (**C**–**E**): Double-blinded estimated H-score based on different ER (**C**), PR (**D**), and HER2 status. Data are presented as graph mean ± SEM. *, *p* < 0.05; **, *p* < 0.01; ***, *p* < 0.001. Statistical significance is calculated by unpaired Student’s *t* test. (**F**,**G**): The relapse-free survival (RFS) of breast cancer patients, based on the RNA level of TMEM97 detected by four probe sets in microarray. The red line represents high expression of TMEM97, while the black line represents low expression. (**F**) The RFS in all types of breast cancer patients, and three out of four probes associate significantly with poor survival. (**G**) The RFS in ER-positive breast cancer patients, all four probes are significantly associated with poor survival.

**Figure 2 cancers-15-05691-f002:**
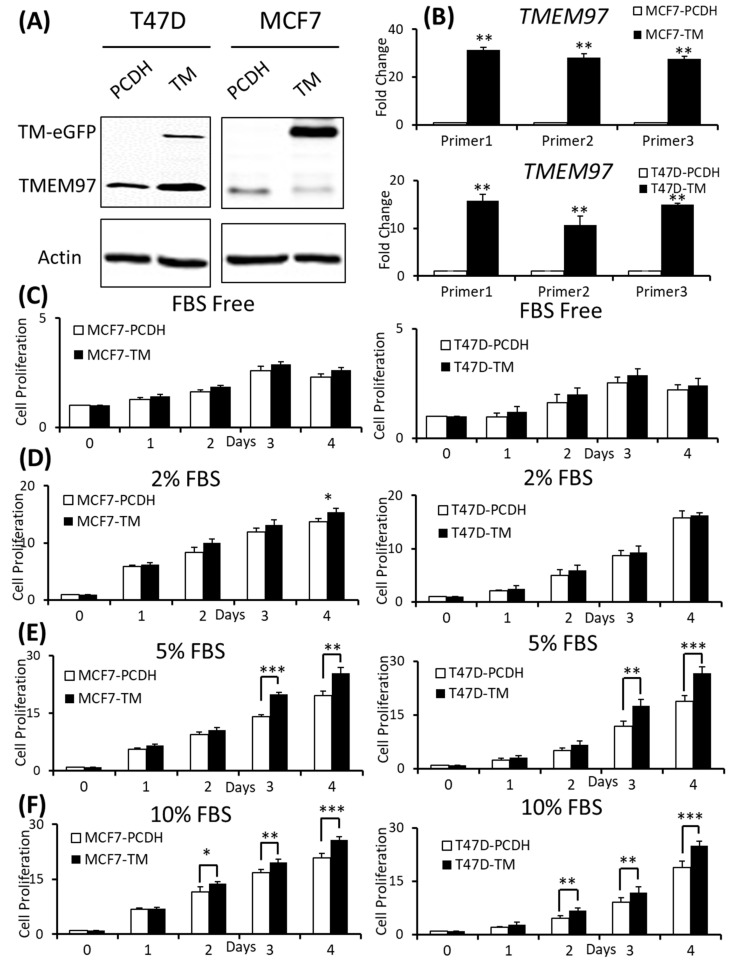
Effects of increased expression of TMEM97 on breast cancer cell growth under different conditions. (**A**): Increased expression of TMEM97 at protein levels in breast cancer cells confirmed by Western blot. (**B**): Increased TMEM97 expression at RNA level confirmed by quantitative RT-PCR. (**C**–**F**): Effects of increased TMEM97 expression on breast cancer cell growth under serum-depleted culture conditions (**C**), 2% FBS (**D**), 5% FBS (**E**), or 10% FBS (**F**). Left, MCF7 control compared with overexpression cells. Right, T47D control compared with overexpression cells. The proliferative capacity was measured by MTS assay and normalized with their respective numbers at Day 0. The results represent the mean ± S.D. of each experiment performed in quadruplicate. *, *p* < 0.05. **, *p* < 0.01. ***, *p* < 0.001 by two-tailed paired Student’s *T* test. The original Western blots file can be found in the Appendix A.

**Figure 3 cancers-15-05691-f003:**
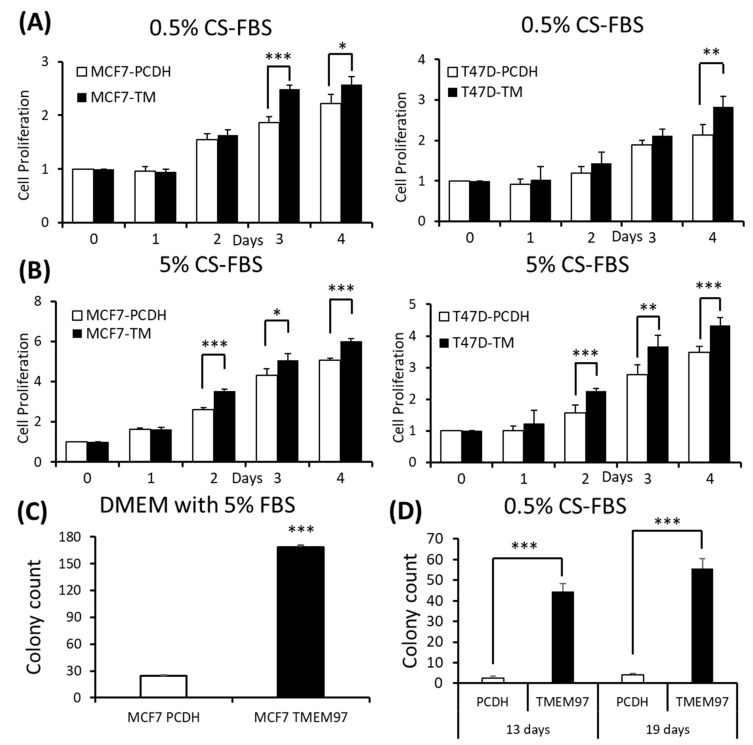
Stimulation of ER-positive breast cancer cell growth and colony formation by TMEM97 under different culture conditions. (**A**,**B**): Effects of increased TMEM97 expression on breast cancer cell growth cultured in phenol red-free and 0.5% charcoal-stripped FBS for 4 days (**A**), or in phenol red-free and 5% charcoal-stripped FBS (**B**). Left, MCF7 control compared with overexpression cells. Right, T47D control compared with overexpression cells. The proliferative capacity was measured by MTS assay and relative Day 0, respectively. (**C**,**D**): Effects of increased TMEM97 expression on breast cancer cell colony formation when cultured in DMEM with 5% FBS after 13 days (**C**) or in phenol red-free DMEM with 0.5% charcoal-stripped FBS after 13 days (**D**, left) and 19 days (**D**, right). The results represent the mean ± S.D. of each experiment performed in quadruplicate. *, *p* < 0.05. **, *p* < 0.01. ***, *p* < 0.001 by two-tailed paired Student’s *T* test.

**Figure 4 cancers-15-05691-f004:**
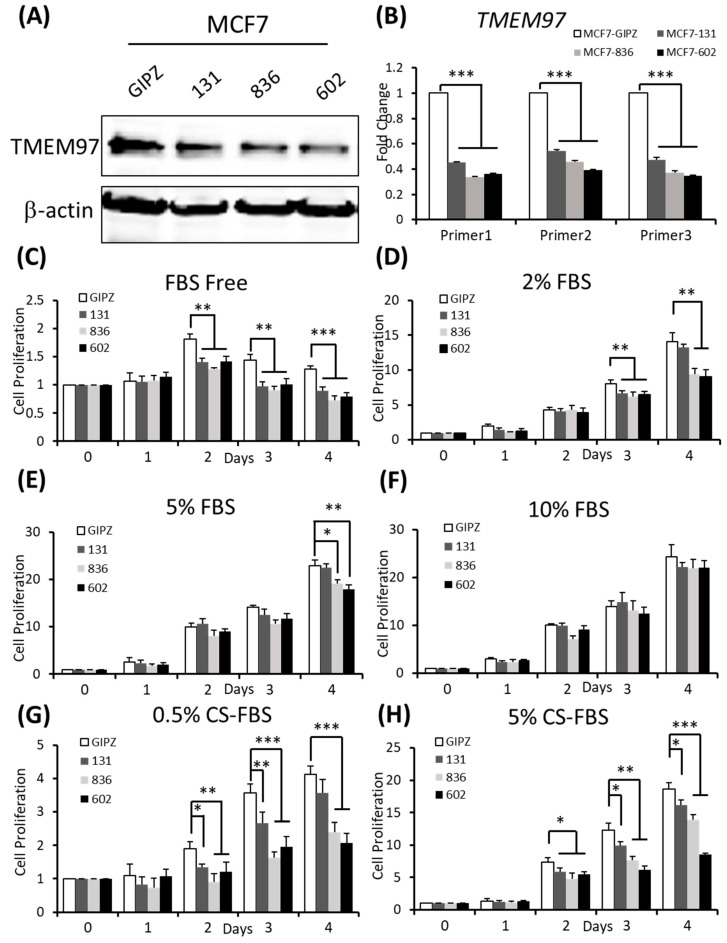
Inhibition of ER-positive breast cancer cell growth by TMEM97 knockdown. (**A**,**B**). Knockdown of TMEM expression by three different shRNAs, as evaluated by immunoblot (**A**) and RT-qPCR with three sets of primers (**B**). (**C**–**H**). Effects of TMEM97 knockdown on cell growth under different cell culture conditions, serum-free media (**C**), media with 2% FBS (**D**), media with 5% FBS (**E**), media with 10% FBS (**F**), phenol red-free media with 0.5% charcoal-stripped FBS (**G**), or phenol red-free media with 5% charcoal-stripped FBS (**H**). The amounts of viable cells at different time points were measured by MTS assays and normalized with the readings at Day 0. The results represent the mean ± S.D. of each experiment performed in quadruplicate. *, *p* < 0.05. **, *p* < 0.01. ***, *p* < 0.001 by two-tailed paired Student’s *t* test. The original Western blots file can be found in the Appendix A.

**Figure 5 cancers-15-05691-f005:**
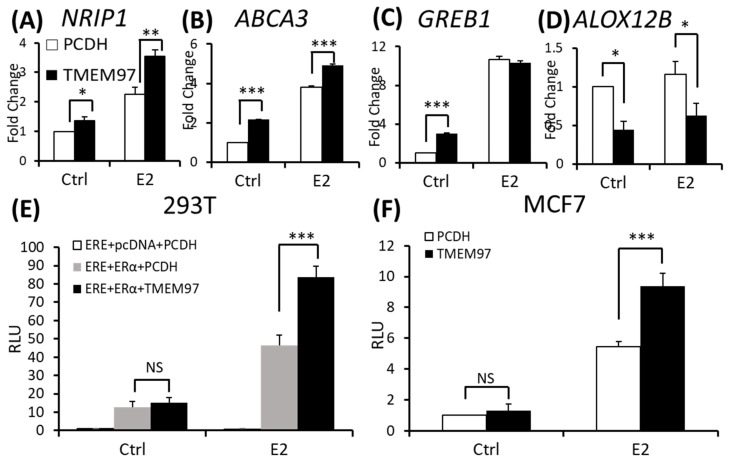
Stimulation of ERα transcriptional activity by TMEM97. (**A**–**D**). Detection of NRIP1 (**A**), ABCA3 (**B**), GREB1 (**C**), and ALOX12B (**D**) mRNA transcripts by RT- qPCR from MCF7 control and TMEM97 overexpression cells grown in phenol red-free/ 0.5% charcoal-stripped FBS medium prior to 5 h 10 µM E2 treatment. (**E**). 293T cells were transit transfected with ERE reporter construct along with ERα, with or without TMEM97eGFP expression vector under full serum culture conditions for 24 h. After being serum and estrogen depleted for another 24 h, cells were subjected to 10 µM E2 treatment for 5 h and harvested for dual luciferase assay. Results show relative luciferase activities. (**F**). Dual luciferase assay for MCF7-TMEM97 overexpression and control cells. The results represent the mean ± S.D. of each experiment performed in quadruplicate. *, *p* < 0.05. **, *p* < 0.01. ***, *p* < 0.001 by two-tailed paired Student’s *t* test.

**Figure 6 cancers-15-05691-f006:**
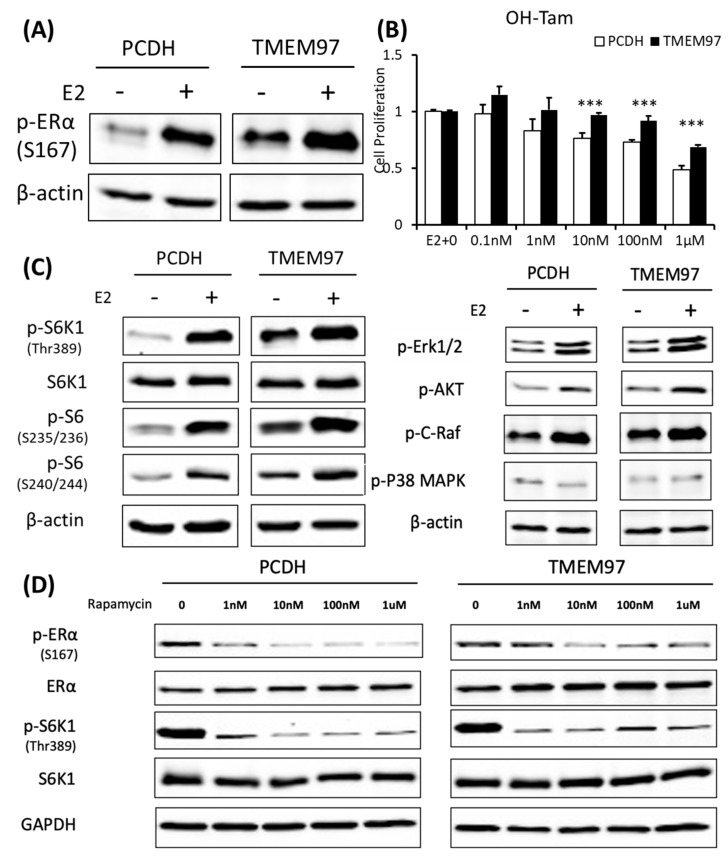
TMEM97 stimulation of ERα activation and tamoxifen resistance can be blocked by mTOR signaling inhibitor. (**A**). Immunoblot detection of p-ERα S167 in MCF7-PCDH/TMEM97 cells with or without 10 µM E2 treatment for 5 h. Cells were first cultured in phenol red-free medium supplied with 0.5% CS-FBS for 24 h and then proceeded to treatment. (**B**). Responsiveness to tamoxifen treatment of PCDH and TMEM97. Cells were serum and estrogen deprived for 24 h prior to 4 days of culture in E2 10 µM and increasing concentrations of OH-Tam as indicated. (**C**). Immunoblot for phosphorylation of mTOR/S6K1 signaling pathway (left) and phosphorylation of other different signaling pathways in MCF7-PCDH and MCF7-TMEM97 cells. (**D**). Effects of mTOR inhibitor rapamycin on PCDH and TMEM97 cell. Cells were cultured in phenol red-free medium supplied with 0.5% CS-FBS for 24 h then treated with rapamycin for 5 h with indicated concentrations and examined for p-ERα (S167) and p-S6K1 (T389) by immunoplot. The results represent the mean ± S.D. of each experiment performed in quadruplicate. ***, *p* < 0.001 by two-tailed paired Student’s T test. The original Western blots file can be found in the Appendix A.

**Figure 7 cancers-15-05691-f007:**
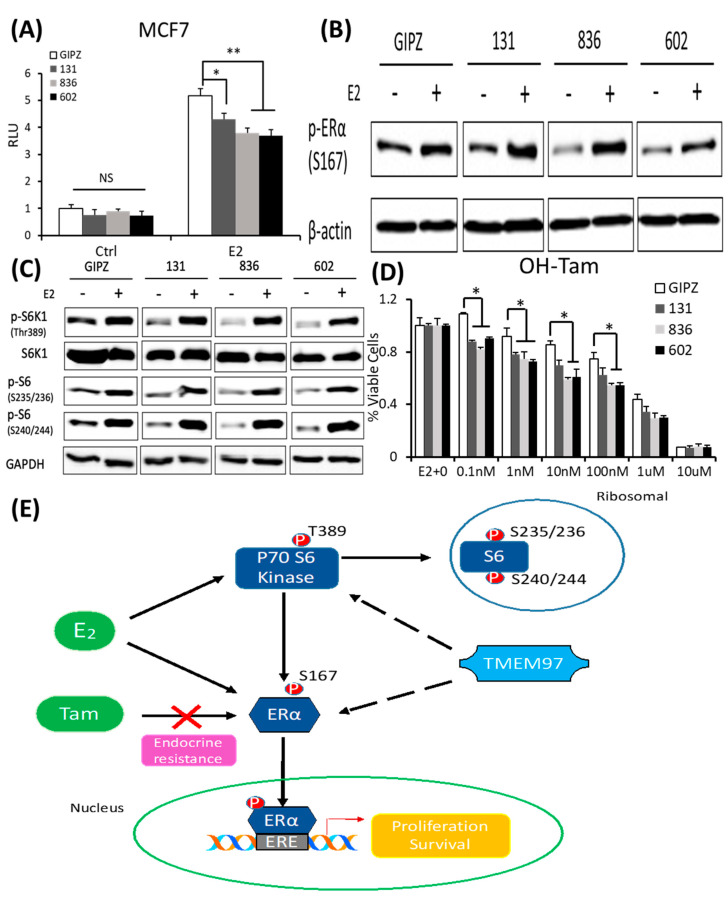
TMEM97 knockdown reduces both ERα and mTOR/S6K1 signaling activity and increases the tamoxifen sensitivity: two mechanisms for TMEM97 to regulate ERα activities. (**A**). Dual luciferase assay for MCF7-GIPZ, 131, 836, and 602 cells. Cells were transiently transfected with ERE reporter construct under full serum culture conditions for 24 h. After being serum and estrogen depleted for another 24 h, cells were subjected to 10 µM E2 treatment for 5 h and harvested for dual luciferase assay. Results show relative luciferase activities. (**B**). Immunoblot detection of p-ERα S167 with or without 10 µM E2 treatment for 5 h under serum- and estrogen-depleted conditions. (**C**). Immunoblot for phosphorylation of mTOR/S6K1 signaling pathway with or without 10 µM E2 treatment for 5 h under serum- and estrogen-depleted conditions. (**D**). Responsiveness to tamoxifen treatment of MCF7-TMEM97 knockdown cells. Cells were serum and estrogen deprived for 24 h prior to 4 days of culture in E2 10µM and increasing concentrations of OH-Tam, as indicated. The proliferative capacity was measured by MTS assay and relative to E2 stimulation alone, respectively. The results represent the mean ± S.D. of each experiment performed in quadruplicate. *, *p* < 0.05. **, *p* < 0.01 by two-tailed paired Student’s *t* test. (**E**). Regulation of ERα by TMEM97. TMEM97 can regulate the activities of ERα through modulating ERα binding to estradiol and binding to responsive elements in the promoters of ER target genes. Another mechanism involved is the TMEM97 modulation of mTOR/S6K1 signaling in the activation of ERα. The original Western blots file can be found in the Appendix A.

**Table 1 cancers-15-05691-t001:** Primers used for gene expression in this study.

Primers	Forward	Reverse
TMEM97 Primer 1	TACCCAGTCGAGTTTAGAAACCT	TGTCATGGTGTGAACAGAGTAGA
TMEM97 Primer 2	ACACCATGACAACCTTAATTCCG	GGGCTCCGCAACATGAAAA
TMEM97 Primer 3	CCCAGCCTGGTTTAAGTCCT	GAAACCACTGGCTTTGGAGA
NRIP1	GCCTGGGGAAGTGTTTGGAT	TGTGCATCTTCTGGCTGTGT
ABCA3	TCTTCGAGCACCCCTTCAAC	GTAGTGTGCCAGCCTTCTGT
GREB1	GGATCTTGTGAGTAGCACTGT	AATCGGTCCACCAATCCCAC
ALOX12B	GAGGAGCATAGAGGCGTTCC	TTCTCAATCAGCACCGGGTC
ESR1	GGGAAGTATGGCTATGGAATCTG	TGGCTGGACACATATAGTCGTT
β-actin	CATGTACGTTGCTATCCAGGC	CTCCTTAATGTCACGCACGAT

**Table 2 cancers-15-05691-t002:** H-score and *p* value calculated for ER, PR, and HER2 status.

			+++/3+	++/2+	+/1+	−/0
ER Status	H-score (mean ± se)		123.16 ± 8.70	77.93 ± 14.97	52.50 ± 10.06	61.21 ± 6.56
	Number of cases (%)		42 (41.2%)	16 (15.7%)	15 (14.7%)	29 (28.4%)
	*p* value	+++		0.014	0.000	0.000
		++			0.265	0.476
		+				0.475
PR Status	H-score (mean ± se)		121.20 ± 8.81	80.42 ± 16.02	72.35 ± 12.41	74.04 ± 7.41
	Number of cases (%)		23 (22.5%)	12 (11.8%)	18 (17.6%)	49 (48.0%)
	*p* value	+++		0.039	0.003	0.000
		++			0.557	0.661
		+				0.776
HER2 Status	H-score (mean ± se)		85.40 ± 10.04	68.33 ± 34.80	55 ± 12.60	81.34 ± 7.17
	Number of cases (%)		31 (30.7%)	3 (3.0%)	8 (7.9%)	59 (58.4%)
	*p* value	3+		0.678	0.054	0.735
		2+			0.744	0.746
		1+				0.059

## Data Availability

Data will be available upon request or deposited in a public database once the manuscript is published.

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
