# Peer review of "TMEM97/Sigma 2 Receptor Increases Estrogen Receptor α Activity in Promoting Breast Cancer Cell Growth"

_cancers, 2023, doi:10.3390/cancers15235691_

Round 1

Reviewer 1 Report

Comments and Suggestions for Authors

This manuscript investigates the role of TMEM97/σ2 receptor in breast cancer, particularly its association with ERα-positive tumors and its impact on cell proliferation, growth, and response to tamoxifen. The authors conclude that TMEM97 is proposed as a novel regulator of ERα, influencing breast cancer cell growth and tamoxifen response. In summary, the study underscores the potential clinical relevance of TMEM97 in ERα-positive breast cancer, shedding light on its role in tumor progression, tamoxifen resistance, and the mTOR signaling pathway. The findings suggest TMEM97 as a candidate for further investigation as a therapeutic target in breast cancer treatment. To enhance the manuscript, consider the following improvements:

1) Can the expression of TMEM97 be influenced by tamoxifen?

2) Have any inhibitors targeting TMEM97 been developed and tested?

3) It would be advantageous for the author to enhance the analysis of signaling pathways. Exploring other components of the mTOR pathway and investigating its crosstalk with ERα signaling could provide a more comprehensive understanding of how TMEM97 influences these pathways.

4) Expanding the study to incorporate a broader spectrum of breast cancer cell lines representing different subtypes would be valuable. This expansion could address questions about the variation in TMEM97 expression by including triple-negative breast cancer cell lines and non-tumorigenic cell lines to validate the results in clinic which TMEM97 expression are dependent on ER and PR expression.

5) Can the proliferation rate of MCF-7 or T47D be assessed under conditions of no fetal bovine serum (FBS) and various concentrations of CS-FBS? Does CS-FBS promote the cell growth?

6) Some figures have low resolution, notably Figure 1F and G. Improving the resolution of these figures, particularly enhancing the clarity of Figure 1F and G, would enhance the overall visual quality of the manuscript.

Author Response

We would like to thank the reviewer for the insightful comments. Below are our point-to-point responses to the concerns raised (Italicized), with corresponding revisions tracked in the manuscript.

Reviewer#1:

Comments and Suggestions for Authors

This manuscript investigates the role of TMEM97/σ2 receptor in breast cancer, particularly its association with ERα-positive tumors and its impact on cell proliferation, growth, and response to tamoxifen. The authors conclude that TMEM97 is proposed as a novel regulator of ERα, influencing breast cancer cell growth and tamoxifen response. In summary, the study underscores the potential clinical relevance of TMEM97 in ERα-positive breast cancer, shedding light on its role in tumor progression, tamoxifen resistance, and the mTOR signaling pathway. The findings suggest TMEM97 as a candidate for further investigation as a therapeutic target in breast cancer treatment. To enhance the manuscript, consider the following improvements:

1) Can the expression of TMEM97 be influenced by tamoxifen?

Responses: The reviewer raised an important question whether TMEM97 expression can be regulated by ERs. We did check the protein level of TMEM97 after tamoxifen treatment of MCF7 cells. We found interesting biphasic responses in terms of TMEM97 protein level after tamoxifen treatment at different doses. Since the observations are preliminary, we did not include the findings in this manuscript.

2) Have any inhibitors targeting TMEM97 been developed and tested?

Responses: Yes, there are a number of compounds that can bind to sigma 1 and sigma 2 receptors with different affinities and specificities. For example, SM21 is considered to be an antagonist of TMEM97/σ2 receptor. Rimcazole is considered to be a low affinity antagonist. However, cautions are needed since many of those compounds can have off-target effects. We have ongoing collaboration with a medicinal chemist to screen and develop compounds with high affinity and selectivity for the TMEM97/σ2 receptors. We have tested the effects of some compounds but the preliminary results are not presented in this paper, due to the consideration of their potential off-target effects.

3) It would be advantageous for the author to enhance the analysis of signaling pathways. Exploring other components of the mTOR pathway and investigating its crosstalk with ERα signaling could provide a more comprehensive understanding of how TMEM97 influences these pathways.

Responses: The reviewer raised a good point regarding expanding the studies to other components of the mTOR pathway and its crosstalk with ERa signaling. The studies are ongoing and the studies will help to elucidate the mechanism of the role of TMEM97 in breast cancers.

4) Expanding the study to incorporate a broader spectrum of breast cancer cell lines representing different subtypes would be valuable. This expansion could address questions about the variation in TMEM97 expression by including triple-negative breast cancer cell lines and non-tumorigenic cell lines to validate the results in clinic which TMEM97 expression are dependent on ER and PR expression.

Responses: The suggestions from the reviewer are well taken. Indeed, there are some reports regarding the involvement of TMEM97 in tumor cell proliferation and survival. In this paper, when noticed a positive correlation of TMEM97 expression and ER/PR expression in clinical tumor specimens, we focused on MCF7 and T47D cell lines with ER/PR expression, since the observations are novel. Nevertheless, it is a good idea to expand the study to a broader spectrum of breast cancer cell lines and tumor specimens in the future studies.

5) Can the proliferation rate of MCF-7 or T47D be assessed under conditions of no fetal bovine serum (FBS) and various concentrations of CS-FBS? Does CS-FBS promote the cell growth?

Responses: Yes, we performed the proliferation study by MTS assay and colony formation assay under different culture conditions to determine whether increased expression of TMEM97 could promote and stimulate breast cancer cell growth and survival.

We found that the TMEM97 overexpression cells (TM) tend to survive better than the control cells (PCDH) under serum depleted condition (no FBS), though the differences were not significant (presented in Figure 2C). When supplied with very low concentration of CS-FBS (0.5%), TMEM97 overexpression cells showed enhanced proliferation both in MCF7 and T47D cells (Figure 3A). With increased concentration of CS-FBS (5%), the difference between the overexpression and the control started to show even at Day2 (Figure 3B). In addition, MCF7 TMEM97 showed significantly increased colony formation when compared with pCDH cells under the condition of 0.5% CS-FBS supplement (Figure 3D). The data suggest that TMEM97 could also promote the cell growth and proliferation under estrogen depleted condition for ER positive breast cancer cells.

CS-FBS does not promote increased cell growth when compared to the normal FBS supplement. The utilization of CS-FBS is to provide a “hormone-free” environment for the cells under investigation.

6) Some figures have low resolution, notably Figure 1F and G. Improving the resolution of these figures, particularly enhancing the clarity of Figure 1F and G, would enhance the overall visual quality of the manuscript.

Responses: We agree with the reviewer that it is difficult to see clearly in Figure 1F and G. The high-resolution figures have been submitted to the journal, and we will modify the manuscript accordingly. The high-resolution figures are also attached below for your review.

Reviewer 2 Report

Comments and Suggestions for Authors

The presented manuscript aimed to explore the potential role of TMEM97/Sigma 2 Receptor in promoting breast cancer cell growth. The manuscript is well-designed, conducted, and effectively written; however, some clarifications are warranted:

  1. The rationale behind selecting MCF7 and T47D cell lines, both ER and PR positive, needs further explanation. Given the reported association of TMEM97 with PR expression, justifying the choice of cell lines expressing either ER or PR exclusively would enhance the informativeness of the study, particularly in understanding the underlying mechanism of action.

  2. Discrepancies are observed between the total number of cases mentioned in the Materials and Methods (100 cases) and Table 2 (102 cases). A clarification regarding this discrepancy is crucial to ensure the accuracy of the reported data and to maintain the integrity of the study.

  3. The manuscript lacks a discussion of potential limitations. Addressing the limitations of the study is essential to provide a more transparent view of the research. Discussing any constraints in the experimental approach or potential biases would contribute to a more comprehensive understanding of the study's scope and implications.

Author Response

We would like to thank the reviewer for the insightful comments. Below are our point-to-point responses to the concerns raised (Italicized), with corresponding revisions tracked in the manuscript.

Reviewer#2:

The presented manuscript aimed to explore the potential role of TMEM97/Sigma 2 Receptor in promoting breast cancer cell growth. The manuscript is well-designed, conducted, and effectively written; however, some clarifications are warranted:

  1. The rationale behind selecting MCF7 and T47D cell lines, both ER and PR positive, needs further explanation. Given the reported association of TMEM97 with PR expression, justifying the choice of cell lines expressing either ER or PR exclusively would enhance the informativeness of the study, particularly in understanding the underlying mechanism of action.

Responses: The reviewer raised an interesting and also important question about using cell lines with ER or PR expression exclusivity. Unfortunately, we don’t have cell lines that express ER or PR exclusively.

Among the breast cancer cell lines from ATCC, MCF7 and T47D are two human breast cancer cell line with ER and PR expression. The MCF7 cell line is one of the most representative cell models for estrogen-dependent breast cancers, in which PR expression is estrogen-dependent. In addition, endogenous PR is too low to mediate progestin response in MCF7 cells in the absence of estrogen (for example, supplying CS-FBS), which makes MCF7 our first choice for investigating the role of TMEM97 in ER positive breast cancer and its potential interaction with ERa.

On the other hand, T47D cell line has high expression of PR, which is commonly used to study progesterone signaling in breast cancer and therapeutic efficacy of anti-progestins. The cell line also expresses ER, so we use T47D to corroborate the findings made in MCF7 cells.

Ideally, in the future we can knock out ER or PR expression in MCF7 and T47D cells and examine the functional relationship of TMEM97 with ER or PR separately.

  1. Discrepancies are observed between the total number of cases mentioned in the Materials and Methods (100 cases) and Table 2 (102 cases). A clarification regarding this discrepancy is crucial to ensure the accuracy of the reported data and to maintain the integrity of the study.

Responses: We thank the reviewer for raising the issue that might be confusing to readers. We revised the manuscript to make it clear and less confusing. Basically, the breast cancer tissue microarray (BC081116b) was purchased from US Biomax, Inc. It contains a total of 110 cases with 100 cases of breast invasive ductal carcinoma and 10 cases of adjacent breast tissue. Each case has pathology information for ER, PR and HER2, provided by the company. Out of these total 110 samples, 102 cases have valid ER status (42 cases of 3+, 16 cases of 2+, 15 cases of 1+, and 29 cases of negative), the rest 8 cases have undetermined ER status. Similarly, 102 cases have valid PR status (23 cases of 3+, 12 cases of 2+, 18 cases of 1+, and 49 cases of negative), and 101 cases have valid HER2 status (31 cases of 3+, 3 cases of 2+, 8 cases of 1+, and 59 cases of negative). We performed IHC staining and calculated H-score for these samples, including both breast cancer and normal breast tissue. Analysis were performed for each pathology status category (102 ER, 102 PR, and 101 HER2). The results are presented in Figure 1 and Table 2.

  1. The manuscript lacks a discussion of potential limitations. Addressing the limitations of the study is essential to provide a more transparent view of the research. Discussing any constraints in the experimental approach or potential biases would contribute to a more comprehensive understanding of the study's scope and implications.

Responses: The reviewer raised a very good point. A paragraph has been added to discuss the limitation of the present studies. We agree with the reviewer that the discussions of limitations can also strengthen the manuscript.
